# Continual Treatment Effect Estimation: Challenges and Opportunities

**Zhixuan Chu[1], Sheng Li[2]**

[1]Ant Group, Hangzhou, China
[2]University of Virginia, Charlottesville, USA
chuzhixuan.czx@alibaba-inc.com, shengli@virginia.edu

## Introduction

A further understanding of cause and effect within observational data is critical across many domains, such as economics, health care, public policy, web mining, online advertising, and marketing campaigns. Although significant advances have been made to overcome the challenges in causal effect estimation with observational data, such as missing counterfactual outcomes and selection bias between treatment and control groups, the existing methods mainly focus on source-specific and stationary observational data. In particular, such learning strategies assume that all observational data are already available during the training phase and from only one source.

Along with the fast-growing segments of industrial applications, this assumption is unsubstantial in practice. Taking Alipay as an example, which is one of the world's largest mobile payment platforms and offers financial services to billion-scale users, a tremendous amount of data containing much privacy-related information is produced daily and collected from different sources. In the following, we further elaborate this problem by two points. The first one is based on the characteristics of observational data, which are incrementally available from non-stationary data distributions. For instance, the electronic financial records for one marketing campaign are growing every day and they may be collected from different cities or even other countries. This characteristic implies that one cannot have access to all observational data at one time point and from one single source. The second reason is based on the realistic consideration of accessibility. For example, when new observational data are available, one may want to refine the previously trained model using both the new data and original data. However, it is likely that the original training data are no longer accessible due to a variety of reasons, e.g., legacy data may be unrecorded, proprietary, the sensitivity of financial data, too large to store, or subject to privacy constraint of personal information (Zhang et al. 2020). This practical concern of accessibility is ubiquitous in various academic and industrial applications. That's what it boiled down to: in the era of big data, we face new challenges in causal inference with observational data. We first presented the contin-

ual causal effect estimation problem in (Chu, Rathbun, and Li 2020a), in which we discussed three desired properties of continual causal inference frameworks, i.e., the **extensibility** for incrementally available observational data, the **adaptability** for various data sources in new domains, and the **accessibility** for an enormous amount of data.

In this position paper, we formally define the problem of continual treatment effect estimation, describe its research challenges, and then present possible solutions to this problem. Moreover, we will discuss future research directions on this topic.

## Related Work

Instead of randomized controlled trials, observational data is obtained by the researcher simply observing the subjects without any interference. It means that the researchers have no control over the treatment assignments, and they just observe the subjects and record data based on observations (Yao et al. 2021). Therefore, from the observational data, directly estimating the treatment effect is challenging due to the missing counterfactual outcomes and the existence of confounders. Recently, powerful machine learning methods such as tree-based methods (Athey and Imbens 2016; Wager and Athey 2018), representation learning (Li and Fu 2017; Shalit, Johansson, and Sontag 2017; Yao et al. 2018; Chu, Rathbun, and Li 2022), meta-learning (Künzel et al. 2019; Nie and Wager 2021), and generative models (Louizos et al. 2017; Yoon, Jordon, and van der Schaar 2018) have achieved prominent progress in treatment effect estimation.

In addition, the combination of causal inference and other research fields also exhibits complementary strengths, such as computer vision (Tang et al. 2020; Liu et al. 2022a), graph learning (Ma et al. 2022; Chu, Rathbun, and Li 2021), natural language processing (Feder et al. 2022; Liu et al. 2022b), and so on. The involved causal analysis helps improve the model's capability of discovering and resolving the underlying system beyond the statistical relationships learned from observational data.

## Problem Definition

Suppose that the observational data contain $n$ units collected from $d$ different domains, and $D_d = \{(x, y, t) | x \in X, y \in Y, t \in T\}$ denotes the dataset collected from the $d$-th domain, which contains $n_d$ units. Let $X$ denote all observed

variables, $Y$ denote the outcomes in the observational data, and $T$ be a binary variable. Let $D_{1:d} = \{D_1, D_2, ..., D_d\}$ be the combination of $d$ datasets, separately collected from $d$ different domains. For $d$ datasets $\{D_1, D_2, ..., D_d\}$, they have the commonly observed variables, but due to the fact that they are collected from different domains, they usually have different distributions with respect to $X$, $Y$, and $T$ in each dataset. Each unit in the observational data received one of two or multiple treatments. Let $t_i$ denote the treatment assignment for unit $i$; $i = 1, ..., n$. For binary treatments, $t_i = 1$ is for the treatment group and $t_i = 0$ for the control group. The outcome for unit $i$ is denoted by $y_t^i$ when treatment $t$ is applied to unit $i$. For observational data, only one of the potential outcomes is observed. The observed outcome is called the factual outcome, and the remaining unobserved potential outcomes are called counterfactual outcomes.

The potential outcome framework has beee widely used for estimating treatment effects (Rubin 1974; Splawa-Neyman, Dabrowska, and Speed 1990). The individual treatment effect (ITE) for unit $i$ is the difference between the potential treated and control outcomes and is defined as:

$$\text{ITE}_i = y_1^i - y_0^i. \tag{1}$$

The average treatment effect (ATE) is the difference between the mean potential treated and control outcomes, which is defined as:

$$\text{ATE} = \frac{1}{n} \sum_{i=1}^{n} (y_1^i - y_0^i). \tag{2}$$

The success of the potential outcome framework is based on the following assumptions (Imbens and Rubin 2015), which ensure that the treatment effect can be identified.

**Assumption 1** *Stable Unit Treatment Value Assumption (SUTVA): The potential outcomes for any unit do not vary with the treatments assigned to other units, and, for each unit, there are no different forms or versions of each treatment level, which lead to different potential outcomes.*

**Assumption 2** *Consistency: The potential outcome of treatment $t$ is equal to the observed outcome if the actual treatment received is $t$.*

**Assumption 3** *Positivity: For any value of $x$, treatment assignment is not deterministic, i.e., $P(T = t | X = x) > 0$, for all $t$ and $x$.*

**Assumption 4** *Ignorability: Given covariates, treatment assignment is independent of the potential outcomes, i.e., $(y_1, y_0) \perp\!\!\!\perp t | x$.*

The goal of ***continual treatment effect estimation*** is to estimate the causal effect of treatments for all available data, including new data $D_d$ and the previous data $D_{1:(d-1)}$, without having access to previous data $D_{1:(d-1)}$.

## Research Challenges

Existing causal effect inference methods, however, are unable to deal with the aforementioned new challenges in continual treatment effect estimation, i.e., extensibility, adaptability, and accessibility. Although it is possible to adapt existing treatment effect estimation methods to cater to these issues, these modified methods still have inevitable defects. Three straightforward adaptation strategies are described as follows:

1. If we directly apply the model previously trained based on original data to new observational data, the performance on new tasks will be very poor due to the domain shift issues among different data sources;

2. Suppose we utilize newly available data to re-train the previously learned model for adapting changes in the data distribution. In that case, old knowledge will be completely or partially overwritten by the new one, which can result in severe performance degradation on old tasks. This is the well-known *catastrophic forgetting* problem (McCloskey and Cohen 1989; French 1999);

3. To overcome the catastrophic forgetting problem, we may rely on the storage of old data and combine the old and new data together, and then re-train the model from scratch. However, this strategy is memory inefficient and time-consuming, and it brings practical concerns such as copyright or privacy issues when storing data for a long time (Samet, Miri, and Granger 2013).

Any of these three strategies, in combination with the existing causal effect inference methods, is deficient.

## Potential Solution

To address the continual treatment effect estimation problem, we propose a **C**ontinual **C**ausal **E**ffect **R**epresentation **L**earning framework (CERL) for estimating causal effect with incrementally available observational data. Instead of having access to all previous observational data, we only store a limited subset of feature representations learned from previous data. Combining selective and balanced representation learning, feature representation distillation, and feature transformation, our framework preserves the knowledge learned from previous data and updates the knowledge by leveraging new data so that it can achieve the continual causal effect estimation for incrementally new data without compromising the estimation capability for previous data. In the following, we will briefly describe the design of our CERL framework. More technical details of CERL are presented in (Chu et al. 2023).

*Framework Overview.* To deal with the incrementally available observational data, the framework of CERL is mainly composed of two components: (1) the baseline causal effect learning model is only for the first available observational data, and thus we don't need to consider the domain shift issue among different data sources. This component is equivalent to the traditional causal effect estimation problem; (2) the continual causal effect learning model is for the sequentially available observational data, where we need to handle more complex issues, such as knowledge transfer, catastrophic forgetting, global representation balance, and memory constraint.

*Baseline Causal Effect Learning Model.* We first train the baseline causal effect learning model for the initial observational dataset and then bring in subsequent datasets. The task on the initial dataset can be converted to a traditional causal effect estimation problem. Owing to the suc-

cess of deep learning for counterfactual inference, we propose to learn the selective and balanced feature representations (Shalit, Johansson, and Sontag 2017; Chu, Rathbun, and Li 2020b) for units in treatment and control groups and then infer the potential outcomes based on learned representation space.

***Sustainability of Model Learning.*** To avoid catastrophic forgetting when learning new data, we propose to preserve a subset of lower-dimensional feature representations rather than all original covariates. We can also adjust the number of preserved feature representations according to the memory constraint.

***Continual Causal Effect Learning.*** We have stored memory and the baseline model. To continually estimate the causal effect for incrementally available observational data, we incorporate feature representation distillation and feature representation transformation to estimate the causal effect for all seen data based on a balanced global feature representation space.

## Research Opportunities

Although significant advances have been made to overcome the challenges in causal effect estimation, real-world applications based on observational data are always very complicated. Unlike source-specific and stationary observational data, most real-world data are incrementally available and from non-stationary data distributions. Significantly, we also face the realistic consideration of accessibility. Our work (Chu, Rathbun, and Li 2020a) might be the first attempt to investigate the continual causal inference problem, and we proposed the corresponding evaluation criteria. However, constructing the comprehensive analytical tools and the theoretical framework derived from this brand-new problem requires non-trivial efforts. Specifically, there are several potential directions for continual causal inference:

- In addition to the distribution shift of the covariates among different domains, there are other potential technical issues for continual effect estimation: for example, perhaps we do not initially observe all the necessary confounding variables and may get access to increasingly more confounders.

- Compared with homogeneous treatment effects (e.g., the magnitude and direction of the treatment effect are the same for all patients, regardless of any other patient characteristics), heterogeneous causal effects could differ for different individuals. This could be another important aspect to consider for the continual treatment effect estimation model.

- The basic assumptions for traditional causal effect estimation may not be completely applicable. New assumptions may be supplemented, or previous assumptions need to be relaxed.

- There exists a natural connection with continual domain adaptation among different times or domains ("continual" causal inference) and between treatment and control groups (continual "causal inference").

- Compared to traditional causal effect estimation tasks based on relatively small datasets, the continual causal

inference method will embrace high-performance computing or cloud computing due to its ambitious objective.

- With the increasing public concern over privacy leakage in data, federated learning, which collaboratively trains the machine learning model without directly sharing the raw data among the data holders, may become a potential solution for continual causal inference.

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
