# OpenReview forum: "Continual Treatment Effect Estimation: Challenges and Opportunities"
_AAAI.org/2023/Bridge/CCBridge — AAAI23 Bridge Continual Causality_

### Official Review · Reviewer_CfXP · 2022-11-24
**Application of continual learning methodologies to causal models**

**Rating:** 6
**Confidence:** 4

**Review:**

The paper starts from the observation that causal methods assume that observational data are entirely present at the beginning. Instead, in many cases data may arrive over time. This may cause problems both in adapting to new domains and to preserve knowledge about previous domains.
The authors discussed a feature replay mechanism to mitigate forgetting, combined with feature distillation from the replay buffer. The idea of feature distillation and feature transformation is not developed in the paper and it is not clear.
However, the application of causal approaches on a stream of data is interesting, since it allows to leverage existing continual learning methodology in combination with causal approaches.

Before delving deeper into the integration of continual and causal methodologies, it would be interesting to check whether causal models actually suffer when learning over time. This is postulated in the paper, but it seems there is no known results supporting the statement.

---

### Official Review · Reviewer_JcCd · 2022-11-30
**Interesting motivating problem; technical problem needs better formulation**

**Rating:** 5
**Confidence:** 5

**Review:**

Summary:

The abstract considers the problem of estimating treatment effects as the distribution of the observational data changes. The research problem that the authors want to focus on is how to use techniques from continual learning to update effect estimation as the observational data changes.

Significance: Overall, the high-level motivation -- to estimate effects as we obtain new observational data -- is compelling for several application areas like medicine, social science and more.

Weaknesses:
The biggest issue I see is that based on standard causal inference assumptions (which the authors appeal to), the causal effect, $\text{ATE} = \mathbb{E}[Y; do(T=1)] - \mathbb{E}[Y; do(T=0)],$
is invariant to changes in the distribution of other covariates $X$. That is, if the causal effect is identified in dataset $D_0$, the first dataset one collects, then there's really no "continual learning" challenge here. The causal effect should be transportable across all other datasets.

Therefore, I think the authors need to be careful about formulating their technical problem. I'm not sure that just distribution shift of the covariates, $P(X)$, is all that interesting. But there are other potentially interesting technical issues for continual effect estimation: for example, perhaps we don't initially observe all the necessary confounding variables but support of $P(X)$ is changing and you may get access to increasingly more confounders.

Another interesting problem could be around heterogenous causal effects, treatment effects that are functions of covariates. These effects wouldn't transport as the covariates' distribution shifts. This could another candidate to consider for continual learning.

In sum, the general question proposed by the authors is interesting, but the technical problem formulation needs to be thought through much more carefully, taking invariances implied by causal models into account.

---

### Official Review · Reviewer_5fu3 · 2022-12-02
**Fits well to the Bridge program**

**Rating:** 8
**Confidence:** 5

**Review:**

This position paper discusses the challenges and opportunities of continual causal inference. To this end, it considers the problem of continual treatment effect estimation using sequential observational data from different sources. It argues that existing methods cannot address and if they are adapted, then there are unwanted defects. To overcome this, the paper presents the Continual Causal Effect Representation Learning (CERL) framework, storing only a limited subset of feature representations learned from previous data (kind of coreset) and conceptualizes it using deep learning for counterfactual inference and maintaining a subset of lower-dimensional feature representation.  Overall, I really like the treatment case made. The problem and the CERL framework fits very well with the bridge program on continual causality. If I may raise a minor issue, I would put the treatment estimation problem into the title since not all kinds of causal models are discussed.

---

### Decision · Program_Chairs · 2022-12-05

**Decision:**

Accept

**Comment:**

Accept - Poster

This paper motivates the problem of continual treatment effect estimation and proposes a continual causal effect representation learning framework. The reviewers agree that the problem is well motivated and a great fit for the bridge program. We suggest that the authors use the additional space in the camera-ready version to integrate the reviewers’ comments and concerns, including the additional discussion/formulation of the technical problem formulation.